# Characteristics of Cancer-Related Fatigue and an Efficient Model to Identify Patients with Gynecological Cancer Seeking Fatigue-Related Management

**DOI:** 10.3390/cancers15072181

**Published:** 2023-04-06

**Authors:** Ying-Wen Wang, Yu-Che Ou, Hao Lin, Kun-Siang Huang, Hung-Chun Fu, Chen-Hsuan Wu, Ying-Yi Chen, Szu-Wei Huang, Hung-Pin Tu, Ching-Chou Tsai

**Affiliations:** 1Department of Obstetrics and Gynecology, Chang Gung Memorial Hospital-Kaohsiung Medical Center, Kaohsiung 833, Taiwan; 2Department of Obstetrics and Gynecology, Chia-Yi Chang Gung Memorial Hospital, Chiayi 613, Taiwan; 3Department of Family Medicine, Chang Gung Memorial Hospital-Kaohsiung Medical Center, Kaohsiung 833, Taiwan; 4Department of Public Health and Environmental Medicine, School of Medicine, College of Medicine, Kaohsiung Medical University, Kaohsiung 807, Taiwan

**Keywords:** cancer-related fatigue, gynecological cancer, quality of life, endometrial cancer, cross-sectional study, fatigue, cervical cancer, ovarian cancer

## Abstract

**Simple Summary:**

Cancer-related fatigue (CRF) is a common somatic discomfort in gynecological cancer patients; however, it is usually overlooked by physicians. We aimed to explore the prevalence of CRF’s and the clinical characteristics of gynecological cancer patients. The results showed that 77.9% of patients had received related CRF management. A five-item predictive model was developed from the identified risk factors contributing to CRF. The risk factors included (1) diagnosis of endometrial/cervical cancer, (2) International Federation of Gynecology and Obstetrics (FIGO) stage >1, (3) Eastern Cooperative Oncology Group (ECOG) status score ≥1, (4) inadequate treatment response, and (5) having received cancer treatment in the past 1 week. The predictive model may help physicians more promptly identify high-risk patients in clinical practice.

**Abstract:**

Cancer-related fatigue (CRF) is the most common somatic discomfort in patients with gynecological cancers. CRF is often overlooked; however, it can impair the patients’ quality of life considerably. This cross-sectional study aimed to identify the clinical characteristics of CRF in gynecological cancer patients. Questionnaires and the International Classification of Diseases 10th Revision (ICD-10) criteria were used to identify CRF. The enrolled patients were further categorized according to the amount of fatigue-related management received. Of the enrolled 190 patients, 40.0% had endometrial cancer, 28.9% had cervical cancer, and 31.1% had ovarian cancer. On the basis of the ICD-10 diagnostic criteria, 42.6% had non-cancer-related fatigue, 10% had CRF, and 51% had BFI-T questionnaire-based fatigue. Moreover, 77.9% of the study cohort had ever received fatigue-related management. Further analysis showed that patients with endometrial/cervical cancer, International Federation of Gynecology and Obstetrics stage >1, Eastern Cooperative Oncology Group performance status score ≥1, inadequate cancer treatment response, and receiving cancer treatment in the past week had a higher probability of receiving more fatigue-related management. The five-item predictive model developed from these factors may help physicians recognize patients seeking more fatigue-related management more efficiently. This is important as they may suffer from a more profound CRF.

## 1. Introduction

According to the Taiwan Cancer Registry Annual Report 2019, endometrial cancer is the fifth most common cancer, while ovarian and cervical cancers hold the seventh and ninth positions, respectively [1], with the epidemiology of gynecological cancer being similar in other developed countries [2]. To achieve a better outcome, patients with gynecological cancer receive multimodal management, including surgery, chemotherapy, radiotherapy, targeted therapy, immunotherapy, or a combination of these [3,4,5,6].

The common side effects in patients receiving cancer treatment are as follows: fatigue, hair loss, nausea, pain, and sleep disturbances [7]. Among these, fatigue is the most common symptom associated with cancer and its treatment. The estimated prevalence of cancer-related fatigue (CRF) ranges from 25% to 99% [8]. CRF may appear to be a trivial symptom; however, it can impact negatively on the patients’ quality of life and make them prone to mental disorders [9,10]. Hofman et al. summarized the impact of CRF, which includes inferior daily living performance, increased mood disturbances, and decreased social activities [11]. Curt et al. surveyed cancer patients and indicated that patients with CRF experience frequent mood disturbances, such as mental exhaustion, diminished interest in daily activities, and frustration with CRF. The study showed that patients with CRF also experienced problems maintaining daily routine activities, such as cleaning the house, climbing stairs, preparing food, exercising, and shopping. In addition, the study indicated that CRF may also result in an economic and occupational impact because 75% of the patients mentioned that they adjusted their employment status because of CRF [12]. Some studies have also indicated that CRF may eventually impact patients’ survival negatively [11].

The definition of CRF proposed by The National Comprehensive Cancer Network (NCCN) practice guidelines is “a persistent subjective sense of tiredness related to cancer or cancer treatment that interferes with usual functioning” [13]. Several validated tools or questionnaires have been developed for the evaluation of CRF [14,15]. A physician can use the diagnostic criteria of the International Classification of Diseases 10th Revision (ICD-10) and a simple rating scale developed in the Common Terminology Criteria for Adverse Events (CTCAE) in clinical practice. Patients can also be evaluated by self-reporting the severity using questionnaires such as Brief Fatigue Inventory (BFI), Multidimensional Fatigue Inventory (MFI), and the Functional Assessment of Chronic Illness Therapy—Fatigue (FACIT-F) [16].

Aside from the CRF resulting from cancer treatment, patients with certain risk factors are prone to CRF. Bower systemic reviewed CRF and indicated that genetic factors, especially those promoting inflammatory processes, pretreatment fatigue, pre-existing mood disturbances, low levels of physical activity, and an elevated body mass index (BMI) are recognized risk factors for CRF. The study also showed a significant correlation between sleep disturbance and CRF; however, the relationship needs further studies to clarify the causality [17]. Additionally, Agarwal et al. indicated that pain, physical function, the ECOG performance status score, tiredness, and albumin levels were associated highly with CRF [18]. Schultz et al. suggested that a higher tumor grade and insomnia also contributed to CRF in breast cancer patients [19], and Hinz et al. indicated that being female, having an advanced tumor stage, with the presence of metastases, and a poor ECOG performance status are risk factors for fatigue in cancer patients [20].

As there is limited information on CRF in patients with gynecological cancer, this study aimed to explore the clinical characteristics of patients with CRF and a more efficient approach to help physicians identify these patients in clinical practice promptly.

## 2. Materials and Methods

### 2.1. Participants

This cross-sectional study was performed at the Department of Obstetrics and Gynecology, Kaohsiung Chang Gung Memorial Hospital (KCGMH), and was approved by the Ethics Committee and the Institutional Review Board of KCGMH (IRB201900669B0). The eligible patients diagnosed with cervical, endometrial, or ovarian cancer receiving cancer-related management or surveillance at the inpatient or outpatient department in KCGMH between 1 June 2019, and 31 August 2020, were enrolled for analysis. The other inclusion criteria were age ≥20 years, the ability to communicate verbally, and the ability to complete the questionnaires. The patients diagnosed with cognitive impairment and those who were unable to complete the questionnaires were excluded. Written informed consent was obtained from all the patients.

### 2.2. Measures

The demographic and clinical information, such as the age, cancer type, International Federation of Gynecology and Obstetrics (FIGO) stage, Eastern Cooperative Oncology Group (ECOG) performance status, and current disease condition, were extracted from electronic medical records. We assessed CRF using the BFI-Taiwan (BFI-T) and ICD-10 diagnostic criteria. The ICD-10 diagnostic criteria refer to a structural interview for CRF, a commonly used physician-oriented diagnostic approach that is used in our institute. The BFI-T is a Chinese-translated version of the original BFI [21]. The BFI-T provides a patient-oriented approach to CRF. The questionnaire can help patients self-report CRF’s severity and illustrate possible functional impairment following CRF. It has three items that are used for evaluating fatigue severity and six for evaluating fatigue-related interference in daily functioning during the past 24 h. All the items are scored from 0 (no fatigue/no interference) to 10 (extreme fatigue/complete interference). The final BFI-T score is calculated from the average of the nine items [22].

The Functional Assessment of Cancer Therapy–General–7 Item Version (FACT-G7) is a shortened, seven-item version of the Functional Assessment of Cancer Therapy–General (FACT-G) [23]. Unlike the BFI-T, which focuses on illustrating the severity of CRF, the FACT-G7 reports primarily on the quality of life associated with CRF. The FACT-G7 questionnaire includes seven common cancer-related symptoms and concerns endorsed by the patients. All the items are scored from 0 (not at all) to 4 (very much). The first to fourth item scores are reverse-calculated (four-item response = item score), but the fifth to seventh item scores are calculated directly (0 + item response = item score). The total FACT-G7 score is the sum of the scores of the seven items. A higher total score indicates a better quality of life [23].

In the cancer symptoms survey, the participants were evaluated for all the associated symptoms they had experienced in the past week, whether related to cancer itself, treatment, or other causes. The score ranged from 0 (no symptoms) to 10 (as bad as one can imagine). Symptoms included pain, fatigue, nausea, vomiting, depression, constipation, hair loss, diarrhea, sleep disturbance, shortness of breath, lack of appetite, weight loss, and nutritional imbalance. The items used in the cancer symptoms survey are modified from the Symptom Distress Scale proposed by McCorkle et al. [24,25].

The different types of fatigue-related management included self-monitoring of fatigue level, energy conservation, and physical activity as well as the use of psychosocial interventions, cognitive behavioral therapy for sleep, nutritional consultations, physically based therapies (such as massage, yoga, acupuncture), astragalus polysaccharide supplements, psychostimulants, steroids, blood transfusion, Chinese medicine, and others [26,27].

### 2.3. Statistics

The data were analyzed using SPSS (version 19.0; IBM Corp., Armonk, NY, USA). The continuous parameters were expressed as the means and standard deviations, and categorical variables were presented as absolute numbers or percentages. The analysis of the continuous variables was performed using the general linear model or the Kruskal–Wallis test, as appropriate. The analysis of the categorical variables was tested using the Fisher’s exact test. The adjusted Ls-Mean was calculated after adjusting for age, cancer type, FIGO stage, ECOG status, and current disease condition using the generalized linear model. Post hoc tests were performed using Dunnett’s multiple comparison test. Dunnett’s test is ideal for examining two or more experimental groups against a single control group [28]. In our study, “patients receiving limited types of fatigue-related management (≤5)” and “patients receiving multiple types of fatigue-related management (>5)” were considered the two experimental groups, and “patients never receiving fatigue-related management” was the control group. Therefore, Dunnett’s test was adopted in the post hoc analysis. The performance of the binary logistic regression model was assessed using the area under the receiver operating characteristic curve. The statistical significance was set at *p* ≤ 0.05.

## 3. Results

One-hundred and ninety patients completed the survey; their demographic characteristics are described in Table 1. The median age of the patients was 56.9 years. Among them, 76 patients (40.0%) had endometrial cancer, 55 (28.9%) had cervical cancer, and 59 (31.1%) had ovarian cancer. One-hundred and seventy-seven patients (93.1%) had ECOG performance status scores of 0 and 1. Most of the patients (97.4%) had the disease under control, while only five patients (2.6%) had a progressive cancer status. When evaluating CRF with different approaches among these patients, the ICD-10 diagnostic criteria identified 19 patients (10%) with CRF and 81 patients (42.6%) with non-cancer-related fatigue. Sixty-one patients (32.1%) had mild CRF, and 36 patients (18.9%) had moderate to severe CRF according to the results of the BFI-T survey. A total of 148 patients (77.9%) had ever received fatigue-related management previously, while 83 patients (43.7%) had received multiple types of fatigue-related management (>5).

When categorizing the patients by the quantity of the types of fatigue-related management that was received, they were classified into three groups (Table 2): 0 (never received fatigue-related management), 1 (received limited types of fatigue-related management (≤5)), and 2 (received multiple types of fatigue-related management (>5)). According to the statistical results, the number of patients who received fatigue-related management was significantly lower in patients who had ovarian cancer, stage I disease, ECOG performance status score 0, controlled current disease condition (complete response or partial response), and not receiving cancer treatment in the last week (*p* < 0.0001).

The results of the cancer symptoms survey, FACT-G7 score, and the number of patients receiving fatigue-related management are shown in Table 3. According to the results, patients not receiving any fatigue-related management tended to have a lower total score (5.74 ± 8.62) in the cancer symptoms survey (*p* < 0.0004). Among all the items in the cancer symptoms survey, fatigue (2.21 ± 2.63) and insomnia (2.07 ± 2.84) were the leading two symptoms in the 190 patients. The patients who did not receive any fatigue-related management tended to have a significantly lower score for fatigue, nausea, vomiting, alopecia, anorexia, and weight loss. Additionally, the FACT-G7 score in patients who did not receive any fatigue-related management was significantly higher (24 ± 3.13 vs. 20.28 ± 4.67 and 20.9 ± 5.62, *p* = 0.0004), which suggested a better quality of life. By setting the cutoff value at 22 for the FACT-G7 score, we were able to distinguish the patients who did not receive fatigue-related management from others (*p* = 0.0002).

On the basis of the results shown in Table 2 and Table 3, the factors that significantly influenced the patients’ amount of receiving fatigue-related management are summarized and highlighted in Table 4. When comparing the patients who received multiple types of fatigue-related management (>5) to patients who never received any fatigue-related management, the following were found to be the predictive performance of individual factors, as illustrated in the area under the receiver operating characteristic curve (AUC): endometrial cancer/cervical cancer (AUC, 0.8), FIGO stage > I (AUC, 0.8), ECOG performance status score ≥ 1 (AUC, 0.73), inadequate treatment response (stable disease or progressive disease; AUC, 0.73), low FACT-G7 score (<22; AUC, 0.65), and received cancer treatment in the past week (AUC, 0.67). When incorporating these six factors into a six-item predictive model, the overall AUC became 0.95. When comparing patients who received limited types of fatigue-related management (≤5) to patients who never received any fatigue-related management, the following were found to be the predictive performance of individual factors: endometrial cancer/cervical cancer (AUC, 0.73), FIGO stage > 1 (AUC, 0.74), ECOG performance status score ≥ 1 (AUC, 0.69), inadequate treatment response (stable disease or progressive disease; AUC, 0.73), low FACT-G7 score (<22; AUC, 0.7), and received cancer treatment in the past week (AUC, 0.68). The predictive performance of the six-item model was 0.91.

The recognition of patients who might seek multiple cancer-related management in clinical practice by physicians is crucial as it indicates that these patients suffer from a profound CRF. Considering that the evaluation by FACT-G7 is still time-consuming in clinical practice, we aimed to identify a more direct and efficient predictive model; therefore, a five-item predictive model was developed without using FACT-G7. With an AUC of 0.9438 in the five-item predictive model (Figure 1), there was no significant statistical difference when comparing the AUC of the six-item and the five-item predictive models for CRF (*p* = 0.5924).

## 4. Discussion

### 4.1. Summary of the Main Results

The discrepancy between the identification of CRF by either the ICD-10 diagnostic criteria or the BFI-T and the number of patients receiving fatigue-related management may be due to an imperfect diagnostic algorithm for CRF. In our study cohort, 48.9% of the patients denied having fatigue according to the the BFI-T questionnaire. When evaluating the same cohort using the ICD-10 diagnostic criteria, 47.4% denied having fatigue. Moreover, if analyzed in the same cohort regarding ever receiving “fatigue-related management,” only 22.1% of patients never received fatigue-related management.

Although there are established diagnostic criteria for CRF according to the ICD-10 or the NCCN guidelines [27], it can be difficult for cancer patients to distinguish CRF from general fatigue, mental discomfort, or sleep disturbances in a precise manner. In our cohort, 52.6% of the patients had ICD-10-diagnosed fatigue when counting non-cancer-related fatigue and CRF together. This ratio is compatible with the proposed ratio from other relevant studies [8]. Regardless of the CRF identified by either diagnostic approach, a physician should provide resources or assistance actively, for those patients who seek multiple types of fatigue-related management (>5), as these patients may suffer from more profound CRF than others.

Poort et al. found no significant differences in the development of CRF between patients with endometrial and ovarian cancer [29], and Sekse et al. did not notice a considerable difference between the gynecological cancer types and the development of CRF [10]. Moreover, there has been limited research on the relationship between cancer type and CRF, and most of the results that have been obtained have been inconclusive.

Our study proposed a five-item predictive model for helping physicians identify those patients that may seek more fatigue-related management. The five-item predictive model demonstrated an outstanding performance with an AUC of 0.9438. Our predictive model may be the first model to identify gynecological cancer patients who require more fatigue-related management. Our study also showed that patients with endometrial and cervical cancers had a significantly higher probability of developing CRF.

### 4.2. The Other Predictive Model

There has been limited research on predictive models for CRF. Among female cancer patients, most of the proposed predictors for CRF were developed for breast cancer patients. Haghighat et al. conducted a prospective study with 112 breast cancer patients, suggesting that depression, anxiety, and pain were significant predictors of CRF [30]. Von Ah et al. proposed that mood disturbance was the most important predictor of CRF before, during, and after adjuvant therapy in breast cancer patients [31]. Schultz et al. indicated that a higher tumor grade and insomnia were predictors of significant CRF in breast cancer patients [19]. On the other hand, a predictive CRF model has also been proposed for patients of all cancer types. Hwang et al. proposed a multidimensional predictive model for CRF, including sadness, drowsiness, pain, poor appetite, irritability, and dyspnea. The AUC of the multidimensional model was 0.88 [32]. Agarwal et al. evaluated 110 patients with advanced cancer and suggested that pain, physical function, ECOG performance status score, tiredness, and albumin levels were independent predictors of CRF [18]. Unlike the above predictors, we proposed a predictive model that incorporated data that can be collected easily from the patients’ clinical information without the use of additional questionnaires., which significantly facilitates the evaluation of CRF in clinical practice.

### 4.3. Results in the Context of Published Literature

Poort et al. reported that 48.0% of patients with gynecological cancer experienced clinically significant fatigue after surgery. Although the prevalence slightly decreased 1 year later, it was still 39.0% [29]. Jewett et al. also illustrated that 53.6% of the patients with gynecological cancer considered fatigue to be the most common problem in their lives [33]. CRF can significantly impair the patients’ quality of life. Liavaag et al. reported that CRF may contribute to more somatic and mental morbidities. These patients also had significantly higher anxiety scores and poorer body image, contributing to an increased use of sedatives and antidepressants [34]. Moreover, there is a connection between CRF and sleep disturbance; however, the exact mechanism or causality remains unknown. While it is easy to infer that patients with CRF can fall asleep easily, several factors can influence their sleep quality negatively. Factors that may impair sleep quality include somatic discomfort, cancer-related treatment, emotional discomfort, and an imbalance between sleep opportunities and sleep ability [35].

### 4.4. Treatment for Cancer-Related Fatigue

Available treatment for CRF includes exercise, psychological management, and the use of pharmacological agents. The patients who perceive that physical activity may increase their discomfort following cancer treatment may be inhibited from regular activity, which is called the “fear of movement,” with the decreased activity further triggering CRF [36]. It has been established that exercise can lead to the most considerable symptom improvement in patients with CRF [37]. Meneses-Echávez et al. reviewed nine studies that addressed the efficacy of exercise for treating CRF and proposed that supervised aerobic exercise was effective in patients with CRF [38]. Cramp et al. also indicate that more significant benefits can be achieved when exercise is done during or after adjuvant cancer therapy. Patients with breast cancer and prostate cancer may benefit from exercise; however, there are no significant benefits for patients with hematological cancer [39]. Carroll et al. showed that hematopoietic agents, corticosteroids, and psychostimulants were commonly used pharmacological agents for patients with CRF [40]. Minton et al. indicate that methylphenidate was the only identified beneficial psychostimulant for patients with CRF in the meta-analysis, resulting in a mean decreased fatigue score of 0.30 (95% CI = −0.54 to −0.05; *p* = 0.02). They also indicated that erythropoietin may benefit CRF patients with anemia. It was found that compared to the use of a placebo, erythropoietin resulted in a mean decreased fatigue score of 0.3 (95% CI = −0.46 to −0.29; *p* = 0.008) [41]. On the basis of current evidence, the efficacy of psychological interventions for CRF is controversial. Jacobsen et al. have suggested that psychological interventions are helpful for patients with CRF [42], while Poort et al. systemically reviewed a wide range of psychosocial interventions for CRF, which included education, cognitive-behavioral therapies (such as changing thoughts and emotions), and supportive group therapies; however, this investigation provided inconclusive results due to study bias, the heterogeneity of the study design, and the small sample size of the enrolled studies [43].

### 4.5. Strengths and Weaknesses

Although the proposed prediction model demonstrated outstanding performance, there were still some limitations to this study. Only 190 patients from one tertiary medical center were enrolled in this study. The limited number of cases may have yielded bias in the analysis and the development of the model for identifying patients seeking more fatigue-related management. Despite these limitations, our study had several strengths. This study focused only on CRF in patients with gynecological cancer. This study identified the risk factors of patients seeking multiple fatigue-related management and proposed a predictive model aimed at helping physicians recognize these patients. This model was efficient and not time-consuming for physicians because the information required was based primarily on patients’ clinical information.

### 4.6. Implications for Practice and Future Research

A prospective study may be necessary in the future to validate the actual performance of the proposed prediction model. If patients have all the risk factors for seeking more fatigue-related management, resources or assistance should be actively provided to them to alleviate their poor quality of life.

## 5. Conclusions

There is a discrepancy among the different diagnostic approaches for CRF. In our study cohort, on the basis of the ICD-10 diagnostic criteria, 42.6% of the patients had non-cancer-related fatigue, and 10% had CRF, while on the basis of the BFI-T questionnaire, 51.0% had fatigue. However, 77.9% of the cohort had ever received fatigue-related management. Further analysis showed that the patients with endometrial/cervical cancer, FIGO stage > I, ECOG performance status score ≥ 1, an inadequate cancer treatment response, and those who received cancer treatment in the past week had a higher probability of receiving more fatigue-related management. The five-item predictive model that was developed from these factors may help physicians recognize patients who might seek more fatigue-related management efficiently, indicating their profound symptoms of CRF, and that management for alleviating CRF should be provided promptly for these patients.

## Figures and Tables

**Figure 1 cancers-15-02181-f001:**
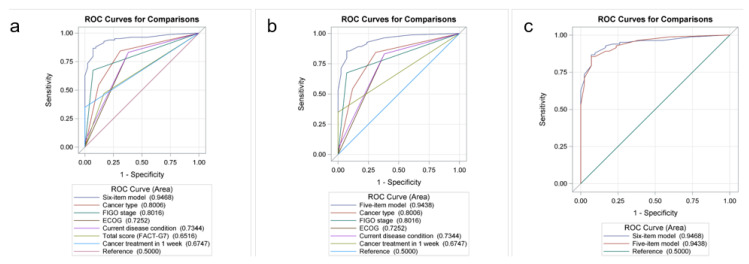
Receiver operating characteristic (ROC) curve and area under the ROC curve (AUC) for predicting the possibility of seeking multiple fatigue-related managements. (**a**) Six-item predictive model. (**b**) Five-item predictive model. (**c**) Comparison between the six-item and five-item models.

**Table 1 cancers-15-02181-t001:** Demographic characteristics of the enrolled patients.

	Patients (*N* = 190)
Age, mean ± SD	56.87 ± 11.89
Cancer type, *n* (%)	
Endometrium cancer	76 (40.0)
Cervical cancer	55 (28.9)
Ovarian cancer	59 (31.1)
FIGO stage, *n* (%)	
I	95 (50.0)
II	29 (15.3)
III	45 (23.7)
IV	21 (11.1)
ECOG performance status, *n* (%)	
0	55 (28.9)
1	122 (64.2)
2	12 (6.3)
3	1 (0.5)
Current disease condition, *n* (%)	
Complete response	48 (25.3)
Partial response	3 (1.6)
Stable disease	134 (70.5)
Progressive disease	5 (2.6)
ICD-10 diagnosed fatigue, *n* (%)	
No fatigue	90 (47.4)
Non-cancer-related fatigue	81 (42.6)
CRF	19 (10.0)
BFI-T questionnaire-based fatigue, *n* (%)	
No: 0	93 (48.9)
Mild: 1–3	61 (32.1)
Moderate to severe: ≥4	36 (18.9)
Fatigue-related management, *n* (%)	
Never	42 (22.1)
Receive limited (≤5) managements	65 (34.2)
Receive multiple (>5) managements	83 (43.7)

SD, standard deviation; FIGO, Federation International of Gynecology and Obstetrics; ECOG, Eastern Cooperative Oncology Group; ICD-10, International Classification of Diseases 10th Revision; CRF, cancer-related fatigue; BFI-T, Brief Fatigue Inventory-Taiwan.

**Table 2 cancers-15-02181-t002:** Clinical characteristics for patients’ frequency of seeking fatigue-related management.

	Fatigue-Related Management			
	0 (*n* = 42)	1 (*n* = 65)	2 (*n* = 83)	*p*-Value
Age, years, mean ± SD	57.96 ± 8.97	58.1 ± 12.96	55.35 ± 12.25	0.3021
≥60, *n* (%)	22 (52.4)	33 (50.8)	52 (62.7)	
<60, *n* (%)	20 (47.6)	32 (49.2)	31 (37.3)	0.2965
Cancer type, *n* (%)				
Endometrial cancer	5 (11.9)	26 (40.0)	45 (54.2)	
Cervical cancer	8 (19.0)	22 (33.8)	25 (30.1)	
Ovarian cancer	29 (69.0)	17 (26.2)	13 (15.7)	<0.0001
FIGO stage, *n* (%)				
I	39 (92.9)	29 (44.6)	27 (32.5)	<0.0001
II	1 (2.4)	15 (23.1)	13 (15.7)	
III	1 (2.4)	15 (23.1)	29 (34.9)	
IV	1 (2.4)	6 (9.2)	14 (16.9)	
ECOG, *n* (%)				
0	26 (61.9)	15 (23.1)	14 (16.9)	<0.0001
1	16 (38.1)	46 (70.8)	60 (72.3)	
2	0 (0.0)	3 (4.6)	9 (10.8)	
3	0 (0.0)	1 (1.5)	0 (0.0)	
Current disease condition, *n* (%)				
Complete response + partial response	26 (61.9)	11 (16.9)	14 (16.9)	<0.0001
Stable disease + progressive disease	16 (38.1)	54 (83.1)	69 (83.1)	
ICD-10-diagnosed fatigue, *n* (%)				
No fatigue	27 (64.3)	29 (44.6)	34 (41.0)	
Non-cancer-related fatigue	13 (31.0)	27 (41.5)	41 (49.4)	
CRF	2 (4.8)	9 (13.8)	8 (9.6)	0.1016
BFI-T questionnaire-based fatigue, *n* (%)				
No: 0	27 (64.3)	28 (43.1)	38 (45.8)	
Mild: 1–3	13 (31.0)	21 (32.3)	27 (32.5)	
Moderate to severe: ≥4	2 (4.8)	16 (24.6)	18 (21.7)	0.0731
FACT-G7, mean ± SD				
Total score	24.00 ± 3.13	20.28 ± 4.67	20.90 ± 5.62	0.0004
Physical well-being	10.38 ± 1.74	9.03 ± 2.23	9.10 ± 2.63	0.0061
Emotional well-being	3.33 ± 0.69	2.58 ± 0.95	2.78 ± 1.12	0.0006
Functional well-being	10.29 ± 1.38	8.66 ± 2.28	9.02 ± 2.63	0.0014
Cancer treatment in recent 1 week, *n* (%)				
No	42 (100.0)	42 (64.6)	54 (65.1)	
Yes	0 (0.0)	23 (35.4)	29 (34.9)	<0.0001

0: never receive fatigue-related management; 1: receive limited (≤5) fatigue-related management; 2: receive multiple (>5) fatigue-related managements. SD, standard deviation; FIGO, International Federation of Gynecology and Obstetrics; ECOG, Eastern Cooperative Oncology Group; ICD-10, International Classification of Diseases 10th Revision; CRF, cancer-related fatigue; BFI-T, Brief Fatigue Inventory-Taiwan; FACT-G7, Functional Assessment of Cancer Therapy–General–7 Item Version. Statistical method: Fischer’s exact test for the comparison of categorical variables; Kruskal–Wallis test for the comparison of continuous variables.

**Table 3 cancers-15-02181-t003:** Association between cancer-related symptoms, FACT-G7, and the frequency of seeking fatigue-related management.

		Fatigue-Related Management				1 vs. 0		2 vs. 0		2 vs. 1	
	Total (*N* = 190)	0 (*n* = 42)	1 (*n* = 65)	2 (*n* = 83)	*p*-Value *	Difference Ls-Mean (95% CI)	*p*-Value	Difference Ls-Mean (95% CI)	*p*-Value	Difference Ls-Mean (95% CI)	*p*-Value
Cancer-related symptoms, mean ± SD	13.04 ± 16.17	5.74 ± 8.62	15.17 ± 14.97	15.07 ± 18.85	0.0004	4.41 (−2.83, 11.64)	0.2816	2.77 (−4.75, 10.28)	0.5929	−1.64 (−7.25, 3.96)	0.7541
Pain	0.86 ± 2.04	0.29 ± 1.29	1.03 ± 2.25	1.02 ± 2.15	0.0471	0.29 (−0.65, 1.24)	0.6812	0.07 (−0.91, 1.06)	0.9765	−0.22 (−0.96, 0.51)	0.7412
Fatigue	2.21 ± 2.63	0.83 ± 1.86	2.69 ± 2.62	2.53 ± 2.76	0.0002	1.56 (0.36, 2.76)	0.0082	1.35 (0.10, 2.59)	0.0317	−0.21 (−1.14, 0.71)	0.8390
Nausea	0.88 ± 2.13	0.07 ± 0.46	0.82 ± 2.04	1.35 ± 2.56	0.0015	0.19 (−0.74, 1.12)	0.8423	0.50 (−0.47, 1.46)	0.3887	0.31 (−0.41, 1.03)	0.5519
Vomiting	0.55 ± 1.70	0.00 ± 0.00	0.46 ± 1.56	0.89 ± 2.12	0.0086	0.11 (−0.65, 0.86)	0.9158	0.37 (−0.42, 1.15)	0.4506	0.26 (−0.33, 0.84)	0.5337
Depression	1.41 ± 2.35	1.00 ± 1.85	1.66 ± 2.66	1.41 ± 2.32	0.8007	0.43 (−0.67, 1.54)	0.5557	0.07 (−1.07, 1.22)	0.9830	−0.36 (−1.22, 0.49)	0.5598
Constipation	1.01 ± 2.19	0.88 ± 2.05	1.49 ± 2.68	0.69 ± 1.75	0.3448	0.32 (−0.72, 1.37)	0.6801	−0.61 (−1.69, 0.48)	0.3340	−0.93 (−1.74, −0.12)	0.0200
Alopecia	1.15 ± 2.46	0.33 ± 1.22	1.88 ± 3.13	0.99 ± 2.19	0.0194	0.94 (−0.19, 2.07)	0.1156	−0.01 (−1.18, 1.17)	0.9998	−0.95 (−1.82, −0.07)	0.0317
Diarrhea	0.53 ± 1.49	0.48 ± 1.40	0.45 ± 1.24	0.63 ± 1.71	0.8727	−0.34 (−1.02, 0.35)	0.4175	−0.32 (−1.03, 0.39)	0.4739	0.02 (−0.51, 0.55)	0.9966
Insomnia	2.07 ± 2.84	1.50 ± 2.42	2.46 ± 3.12	2.05 ± 2.78	0.1922	0.65 (−0.71, 2.00)	0.4359	0.19 (−1.22, 1.59)	0.9282	−0.46 (−1.51, 0.59)	0.5348
Shortness of breath	0.62 ± 1.59	0.33 ± 1.18	0.45 ± 1.20	0.90 ± 1.96	0.2743	−0.03 (−0.78, 0.73)	0.9934	0.46 (−0.32, 1.25)	0.3013	0.49 (−0.09, 1.08)	0.1146
Anorexia	0.92 ± 2.16	0.02 ± 0.15	1.08 ± 2.41	1.24 ± 2.38	0.0012	0.38 (−0.60, 1.37)	0.5588	0.34 (−0.68, 1.36)	0.6472	−0.04 (−0.80, 0.72)	0.9887
Weight loss	0.41 ± 1.53	0.00 ± 0.00	0.25 ± 0.83	0.73 ± 2.15	0.0247	−0.12 (−0.81, 0.58)	0.8907	0.3 (−0.42, 1.02)	0.5240	0.42 (−0.12, 0.95)	0.1609
Nutrition imbalance	0.44 ± 1.65	0.00 ± 0.00	0.46 ± 1.74	0.64 ± 1.95	0.0506	0.01 (−0.77, 0.79)	0.9991	0.06 (−0.75, 0.87)	0.9773	0.05 (−0.56, 0.65)	0.9794
FACT-G7, mean ± SD											
Total score	21.37 ± 5.03	24.00 ± 3.13	20.28 ± 4.67	20.90 ± 5.62	0.0004	−2.25 (−4.48, −0.03)	0.0467	−1.26 (−3.57, 1.06)	0.3537	1.00 (−0.73, 2.72)	0.3446
Physical well-being	9.36 ± 2.37	10.38 ± 1.74	9.03 ± 2.23	9.10 ± 2.63	0.0061	−0.67 (−1.77, 0.43)	0.2848	−0.41 (−1.56, 0.73)	0.6035	0.25 (−0.60, 1.11)	0.7465
Emotional well-being	2.84 ± 1.01	3.33 ± 0.69	2.58 ± 0.95	2.78 ± 1.12	0.0006	−0.62 (−1.07, −0.17)	0.0053	−0.38 (−0.85, 0.09)	0.1210	0.23 (−0.12, 0.58)	0.2451
Functional well-being	9.18 ± 2.35	10.29 ± 1.38	8.66 ± 2.28	9.02 ± 2.63	0.0014	−0.97 (−2.00, 0.06)	0.0668	−0.46 (−1.53, 0.61)	0.5010	0.51 (−0.29, 1.31)	0.2740

0: never receive fatigue-related management; 1: receive limited (≤5) fatigue-related management; 2: receive multiple (>5) fatigue-related managements. SD, standard deviation; Ls-Mean, least squares mean; CI, confidence interval; FACT-G7, Functional Assessment of Cancer Therapy–General–7 Item Version. * *p*-values were calculated using the general linear model or Kruskal–Wallis test, as appropriate. Adjusted Ls-Mean was calculated after adjusting for age group, cancer type, stage, ECOG, and current disease condition using the generalized linear model. Post hoc tests were performed using Dunnett’s multiple comparison test.

**Table 4 cancers-15-02181-t004:** Predictors for patients seeking multiple types of fatigue-related management (>5).

	Fatigue-Related Management				1 vs. 0	2 vs. 0	1 vs. 0		2 vs. 0	
	0 (*n* = 42)	1 (*n* = 65)	2 (*n* = 83)	*p*-Value	AUC (95% CI)	AUC (95% CI)	Adjusted OR (95% CI)	*p*-Value	Adjusted OR (95% CI)	*p*-Value
Cancer type, *n* (%)					0.73 (0.64–0.82)	0.80 (0.72–0.88)				
Ovarian cancer	29 (69.0)	17 (26.2)	13 (15.7)	<0.0001			1.00		1.00	
Cervical cancer	8 (19.0)	22 (33.8)	25 (30.1)				2.99 (0.85–10.49)	0.7327	3.64 (1.03–12.94)	0.9058
Endometrial cancer	5 (11.9)	26 (40.0)	45 (54.2)				5.85 (1.52–22.51)	0.0610	11.49 (3.04–43.48)	0.0049
FIGO Stage, *n* (%)					0.74 (0.67–0.81)	0.80 (0.74–0.87)				
I	39 (92.9)	29 (44.6)	27 (32.5)				1.00		1.00	
>I	3 (7.1)	36 (55.4)	56 (67.5)	<0.0001			10.92 (2.64–45.16)	0.0010	15.42 (3.80–62.65)	0.0001
ECOG performance status, *n* (%)					0.69 (0.60–0.78)	0.73 (0.64–0.81)				
0	26 (61.9)	15 (23.1)	14 (16.9)				1.00		1.00	
≥1	16 (38.1)	50 (76.9)	69 (83.1)	<0.0001			1.00 (0.18–5.50)	0.9987	2.58 (0.44–15.15)	0.2934
Current disease condition					0.73 (0.64–0.81)	0.73 (0.65–0.82)				
Complete response + partial response	26 (61.9)	11 (16.9)	14 (16.9)				1.00		1.00	
Stable disease + progressive disease	16 (38.1)	54 (83.1)	69 (83.1)	<0.0001			4.59 (0.82–25.82)	0.9844	1.89 (0.32–11.17)	0.9788
Total score (FACT-G7)					0.70 (0.62–0.78)	0.65 (0.57–0.73)				
≥22	35 (83.3)	28 (43.1)	44 (53.0)				1.00		1.00	
<22	7 (16.7)	37 (56.9)	39 (47.0)	0.0002			9.09 (2.82–29.28)	0.0002	5.63 (1.70–18.64)	0.0047
Cancer treatment in recent 1 week					0.68 (0.62–0.74)	0.67 (0.62–0.73)				
No	42 (100.0)	42 (64.6)	54 (65.1)				1.00		1.00	
Yes *	0 (0.0) †	23 (35.4)	29 (34.9)	<0.0001						
AUC (95% CI): combined factors ‡					0.91 (0.86–0.97)	0.95 (0.91–0.98)				

FIGO, International Federation of Gynecology and Obstetrics; ECOG, Eastern Cooperative Oncology Group; FACT-G7, Functional Assessment of Cancer Therapy–General–7 Item Version; AUC, area under the receiver operating characteristic curve; CI, confidence interval; OR, odds ratio. The adjusted OR with 95% CI was calculated using a multiple multinomial logistic regression model. * Cell contains 0 by trend analysis (as a continuous variable per category). † Cells containing 0, not calculated. ‡ The performance of the binary logistic regression model was assessed using the AUC.

## Data Availability

The datasets used in the current study are available from the corresponding author upon reasonable request.

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
