# Peer review of "Characteristics of Cancer-Related Fatigue and an Efficient Model to Identify Patients with Gynecological Cancer Seeking Fatigue-Related Management"

_cancers, 2023, doi:10.3390/cancers15072181_

Round 1

Reviewer 1 Report

In my opinion, the Introduction should be broader. There is a need for more literature data on fatigue in malignant neoplasms, and above all on preventive methods against fatigue and pain, such as physical activity. There is nothing written about sport in oncology, current recommendations, and above all about the phenomenon of kinesiophobia, fear of movement during and after oncological treatment, and about insufficient management of physical activity of patients in the hospital. Oncological patients are not sufficiently educated that it is possible to live actively with cancer.

Author Response

Response to the reviewers’ comments:

Reviewer #1's comments:

Point 1: In my opinion, the Introduction should be broader. There is a need for more literature data on fatigue in malignant neoplasms, and above all on preventive methods against fatigue and pain, such as physical activity. There is nothing written about sport in oncology, current recommendations, and above all about the phenomenon of kinesiophobia, fear of movement during and after oncological treatment, and about insufficient management of physical activity of patients in the hospital. Oncological patients are not sufficiently educated that it is possible to live actively with cancer.

Response 1: Thank you for your comments. We have added more information in the Introduction and Discussion sections. We have also added information about the treatment for CRF in the discussion section (lines 318–344)

Reviewer 2 Report

In this cross-sectional study, Ying-Wen Wang et al. enrolled 190 patients in the analysis based on a very strict exclusion and inclusion criteria, not only the basic info (age, cancer types and status etc.) was taken, the patients also completed multiple evaluation forms (ICD-10, FACT-G7, cancer symptoms survey etc.) of their overall wellness but not just cancer condition. The authors did further analysis on the Fatigue-related Management, which could be a more accurate indicator reflecting the patients’ fatigue level than the validated tools or questionnaires (which proved to underestimate the fatigue level later). Cancer-related fatigue (CRF) is a very underestimated and overlooked health issue in cancer treatment plan, ‘it can negatively impact patients’ daily functioning, diminish their quality of life, and make them prone to mental discomfort’. The mental wellness of cancer patients is associated with overall clinical outcome, which is supposed to get more attention in the customized individual cancer treatment.

Throughout the study, the authors eventually built up a 5-item prediction model, with higher sensitivity to detect the CRF than the validated ICD-10 or BFI-T, which underestimated the prevalence of fatigue in patients with gynecological cancers. The tool will be helpful to assist physicians recognize patients suffering from CRF and interfere with it, for the better clinical outcome. Ying-Wen Wang et al. did a cross-sectional study that is of significant clinical relevance.

I really appreciate the efforts that Ying-Wen Wang et al. made. They also did a very scientific and deep discussion. I suggest this study be accepted.

Author Response

Response to the reviewers’ comments:

Reviewer #2's comments:

Point 1: In this cross-sectional study, Ying-Wen Wang et al. enrolled 190 patients in the analysis based on a very strict exclusion and inclusion criteria, not only the basic info (age, cancer types and status etc.) was taken, the patients also completed multiple evaluation forms (ICD-10, FACT-G7, cancer symptoms survey etc.) of their overall wellness but not just cancer condition. The authors did further analysis on the Fatigue-related Management, which could be a more accurate indicator reflecting the patients’ fatigue level than the validated tools or questionnaires (which proved to underestimate the fatigue level later). Cancer-related fatigue (CRF) is a very underestimated and overlooked health issue in cancer treatment plan, ‘it can negatively impact patients’ daily functioning, diminish their quality of life, and make them prone to mental discomfort’. The mental wellness of cancer patients is associated with overall clinical outcome, which is supposed to get more attention in the customized individual cancer treatment. Throughout the study, the authors eventually built up a 5-item prediction model, with higher sensitivity to detect the CRF than the validated ICD-10 or BFI-T, which underestimated the prevalence of fatigue in patients with gynecological cancers. The tool will be helpful to assist physicians recognize patients suffering from CRF and interfere with it, for the better clinical outcome. Ying-Wen Wang et al. did a cross-sectional study that is of significant clinical relevance. I really appreciate the efforts that Ying-Wen Wang et al. made. They also did a very scientific and deep discussion. I suggest this study be accepted.

Response 1: Thank you for the positive feedback; we hope that the five-item predictive model helps physicians identify patients with CRF effectively and provides prompt treatment for the patients.

Reviewer 3 Report

Authors submitted a paper that aims to propose a predictive model to predict cancer-related fatigue in gynecological cancer patients.

The idea is of interest for clinicians and patients and could contribute to better detect and take care of fatigue in this population of patients.

Below are listed recommendations to improve the manuscript:

1- Simple summary

- Please describe what is FIGO and ECOG

- The last sentence is too long and unclear. Please modify.

2- Abstract

- Please specify that half of patients had no fatigue complaints and that fatigue reported in the other half was mild to moderate. This has implications for the rest of the results.

3- Introduction

- The introduction is clear, although short. Still, I miss crucial information. 

- Please add information about the factors related to fatigue, leading to a clearer overview of the factors that will be used in the predictive model and why. Such information are notably given in the discussion.

- Please add information related to the various approaches to report fatigue. Why focusing on the ICD and the BFI? Others questionnaires are available and mostly used such FACIT-F, the MFI or the EORTC.

Please consider including the following articles in your introduction and/or discussion:

Friedrich et al., 2019, Quality of life ; Hinz et al., 2020, Supportive Care in Cancer ; Susanne et al., 2019, Supportive Care in Cancer; Richardson, 1998, Supportive Care in Cancer; Stone and Minton, 2008, Europ J Cancer; Luckett et al., 2010, Int. J. Gynecol. Cancer; Seyidova-Khoshknabi,  Mellar P DavisDeclan Walsh, 2011; Minton et al., 2013, Cancer; Al Maqbali, 2017, CROH.

4- Methods

- Please justify the use of both instruments to measure fatigue.

- Please justify the use of the Fact-G.

- Please specify what is the cancer symptoms survey and add a reference. Seemly for the following paragraph.

- Please give more details on the area under curve analysis and its interpretation. Also, please add in the tables which statistical tests was used. Why using the Dunnett test ?

5- Results

- Please add information related to treatments and the time since the latest one.

- Half of patients did not report fatigue, in contrast with information given in the introduction/discussion. This should be discussed and suggest that the instruments used were not appropriate to quantify fatigue in those patients.

- Why information is given about treatment in recent 1 week? Why one week and not another time frame ?

- In Table 3, I can not understand why several variables are in bold, while others not.

- Table 4 is not easy to read in its current form. Please update.

- Results related to roC curve are not clear enough and the associated figures need a clearer legend.

6- The discussion needs to be improved and to include discussion in regards of others recent paper published (see suggested references above but also others). Other papers dedicated to predict fatigue have been published and their results could inform about the current ones.

Author Response

Response to the reviewers’ comments:

Reviewer #3's comments:

Point 1: Authors submitted a paper that aims to propose a predictive model to predict cancer-related fatigue in gynecological cancer patients. The idea is of interest for clinicians and patients and could contribute to better detect and take care of fatigue in this population of patients.

Response 1: Thank you for the positive feedback; we hope that the five-item predictive model helps physicians identify patients with CRF effectively and provides prompt treatment for the patients.

Below are listed recommendations to improve the manuscript:

(Simple summary)

Point 1: Please describe what is FIGO and ECOG. The last sentence is too long and unclear. Please modify.

Response 1: We have added explanations for FIGO and ECOG. We also refined the content in the simple summary and hope that it will now be easier for the readers to follow.

(Abstract)

Point 2: Please specify that half of patients had no fatigue complaints and that fatigue reported in the other half was mild to moderate. This has implications for the rest of the results.

Response 2: We agree that there are discrepancies between the different approaches for identifying patients with CRF. In our study cohort, based on the ICD-10 diagnostic criteria, 42.6% of the patients had non-cancer-related fatigue, and 10% had CRF while based on the BFI-T questionnaire, 51.0% had fatigue. However, 77.9% of the cohort had ever received fatigue-related management. Further analysis showed that the patients with endometrial/cervical cancer, FIGO stage > I, ECOG performance status score ≥ 1, an inadequate cancer treatment response, and those who received cancer treatment in the past week had a higher probability of receiving more fatigue-related management. The five-item predictive model that was developed from these factors may help physicians recognize patients who might seek more fatigue-related management efficiently, indicating their profound symptoms of CRF, and that management for alleviating CRF should be provided promptly for these patients.

(Introduction)

Point 3: The introduction is clear, although short. Still, I miss crucial information.

- Please add information about the factors related to fatigue, leading to a clearer overview of the factors that will be used in the predictive model and why. Such information are notably given in the discussion.

Response 3: Thank you for your suggestion. The discussion of risk factors was scattered throughout our original manuscript's Introduction and Discussion sections. We have reconstructed the Introduction section (lines 77–89), and we hope that the re-arrangement will now make it easier for the readers to follow.

We have also included additional information in the Introduction section to make it more comprehensive.

Point 4: Please add information related to the various approaches to report fatigue. Why focusing on the ICD and the BFI? Others questionnaires are available and mostly used such FACIT-F, the MFI or the EORTC.

Response 4: We have added relevant information in the Introduction section (lines 67–76) and the Materials and Methods section (lines 105–117). The ICD diagnostic criteria and the rating scale of the EORTC are commonly used tools by physicians in clinical practice and can be considered as physician-orientated approaches for CRF. In contrast, the questionnaires such as the FACIT-F, MFI, and BFI can be regarded as patient-orientated evaluations for CRF. The ICD-10 diagnostic criteria provides a more comprehensive evaluation than the EORCT rating scale; therefore, we justified the ICD-10 diagnostic criteria as the representative physician-oriented evaluation for CRF. On the contrary, the BFI is a commonly used questionnaire in Taiwan; therefore, we adopted the BFI-T as the representative patient-oriented evaluation for CRF.

Point 5: Please consider including the following articles in your introduction and/or discussion: Friedrich et al., 2019, Quality of life ; Hinz et al., 2020, Supportive Care in Cancer ; Susanne et al., 2019, Supportive Care in Cancer; Richardson, 1998, Supportive Care in Cancer; Stone and Minton, 2008, Europ J Cancer; Luckett et al., 2010, Int. J. Gynecol. Cancer; Seyidova-Khoshknabi,  Mellar P Davis, Declan Walsh, 2011; Minton et al., 2013, Cancer; Al Maqbali, 2017, CROH.

Response 5: Thank you for providing these relevant articles. We have added most of them in our Discussion and to our References.

(Methods)

Point 6: Please justify the use of both instruments to measure fatigue. Please justify the use of the Fact-G.

Response 6: We have added more information about how we justified the diagnostic approaches to CRF in the Materials and Methods section (lines 107–119). We also modified the paragraph regarding the FACT-G7 and how we adjusted its use in our study in the Materials and Methods section (lines 120–129).

Point 7: Please specify what is the cancer symptoms survey and add a reference. Seemly for the following paragraph.

Response 7: We have added references for the cancer symptoms survey in the Materials and Methods section (lines 130–136).

Point 8: Please give more details on the area under curve analysis and its interpretation. Also, please add in the tables which statistical tests was used. Why using the Dunnett test?

Response 8: We have refined the interpretation of the AUC in the Results section (lines 217–234) and hope the data are easier for the readers to understand. We have also explained Dunnett's test in greater detail and how we justified its use in the analysis in the Materials and Methods (line 151–159).

(Results)

Point 9: Please add information related to treatments and the time since the latest one.

Response 9: In our analysis, we only surveyed the types and quantities of fatigue-related management received by patients with fatigue. The different types of fatigue-related management included self-monitoring of fatigue level, energy conservation, and physical activity as well as the use of psychosocial interventions, cognitive behavioral therapy for sleep, nutritional consultations, physically based therapies (such as massage, yoga, acupuncture), astragalus polysaccharide supplements, psychostimulants, steroids, blood transfusion, Chinese medicine, and others (lines 137–142). However, we did not have detailed information about their duration and timing. In addition, we also added more information about CRF treatment in the Discussion section (lines 318-344).  

Point 10: Half of patients did not report fatigue, in contrast with information given in the introduction/discussion. This should be discussed and suggest that the instruments used were not appropriate to quantify fatigue in those patients.

Response 10: We agree that there is a discrepancy in identifying patients with CRF according to the different diagnostic approaches used. We elaborated on the discrepancy in the Discussion section (lines 257–272).

Point 11: Why information is given about treatment in recent 1 week? Why one week and not another time frame?

Response 11: Many patients who were receiving ongoing cancer treatment, such as chemotherapy and radiotherapy, were included in our study. These patients attended the out-patient department within 2 weeks after their cancer treatment to monitor any adverse events. This may explain why they could better recall the timing of CRF following the cancer treatment.

Point 12: In Table 3, I can not understand why several variables are in bold, while others not.

Response 12: The variables were written in bold remind us to report those important results in the Results section and to discuss them in the Discussion section when writing the manuscript. We have removed them.

Point 13: Table 4 is not easy to read in its current form. Please update.

Response 13: We have modified Table 4 and the Results section (lines 217–234), and we hope the adjustment will make it easier for the readers to follow.

Point 14: Results related to roC curve are not clear enough and the associated figures need a clearer legend.

Response 14: We have upgraded the resolution of the figure and hope it is clearer. We also modified the results with regard to the ROC and hope the explanation makes it easier for the readers to follow (lines 217–249).

Point 15: The discussion needs to be improved and to include discussion in regards of others recent paper published (see suggested references above but also others). Other papers dedicated to predict fatigue have been published and their results could inform about the current ones.

Response 15: Thank you for your comments. We have added more information and reconstructed the Discussion, and we hope that it will provide a more comprehensive review of the relevant issues raised in our study.

Round 2

Reviewer 3 Report

The authors have successfully adress my comments, thank you for providing this revised version.